# An Update on the Therapeutic Anticancer Potential of *Ocimum sanctum* L.: “Elixir of Life”

**DOI:** 10.3390/molecules28031193

**Published:** 2023-01-25

**Authors:** Mohammad Raghibul Hasan, Bader Saud Alotaibi, Ziyad Mohammed Althafar, Ahmed Hussain Mujamammi, Jafar Jameela

**Affiliations:** 1Department of Clinical Laboratory Sciences, College of Applied Medical Sciences, Shaqra University, Alquwayiyah 11971, Saudi Arabia; 2Clinical Biochemistry Unit, Department of Pathology, College of Medicine, King Saud University, Riyadh 11461, Saudi Arabia

**Keywords:** tulsi, *Ocimum sanctum*, eugenol, cancer, anti-inflammatory, antioxidant

## Abstract

In most cases, cancer develops due to abnormal cell growth and subsequent tumour formation. Due to significant constraints with current treatments, natural compounds are being explored as potential alternatives. There are now around 30 natural compounds under clinical trials for the treatment of cancer. Tulsi, or Holy Basil, of the genus Ocimum, is one of the most widely available and cost-effective medicinal plants. In India, the tulsi plant has deep religious and medicinal significance. Tulsi essential oil contains a valuable source of bioactive compounds, such as camphor, eucalyptol, eugenol, alpha-bisabolene, beta-bisabolene, and beta-caryophyllene. These compounds are proposed to be responsible for the antimicrobial properties of the leaf extracts. The anticancer effects of tulsi (*Ocimum sanctum* L.) have earned it the title of “queen of herbs” and “Elixir of Life” in Ayurvedic treatment. Tulsi leaves, which have high concentrations of eugenol, have been shown to have anticancer properties. In a various cancers, eugenol exerts its antitumour effects through a number of different mechanisms. In light of this, the current review focuses on the anticancer benefits of tulsi and its primary phytoconstituent, eugenol, as apotential therapeutic agent against a wide range of cancer types. In recent years, tulsi has gained popularity due to its anticancer properties. In ongoing clinical trials, a number of tulsi plant compounds are being evaluated for their potential anticancer effects. This article discusses anticancer, chemopreventive, and antioxidant effects of tulsi.

## 1. Introduction

Throughout our lives, all of the cells in our bodies undergo mutations, and occasionally, a bad combination of mutations leads to the onset of cancer [1]. The word “cancer” is used to describe a group of diseases that can affect any part of the body. In 2020, there were 19.29 million new cases of cancer worldwide, which killed 9.96 million people. Cancer was the leading cause of death worldwide in 2020. Globally, the incidence and death rates for cancer have been increasing quickly; this is due to the population’s ageing and expansion, as well as variations in the frequency and distribution of the key risk factors for cancer, many of which are linked to socioeconomic development.

Natural substances have long been a rich source of beneficial therapeutic ingredients. More than 50% of the medications that are now on the market contain natural substances or molecules that are connected to them. The percentage of natural products in cancer therapies is around 60%. More than 30 natural substances are now being studied in various stages of clinical trials for the treatment of various cancers [2].

Holy Basil (tulsi in Sanskrit) is an aromatic herb (Figure 1A) from the Lamiaceae family that has been used in Ayurvedic medicine for over 3000 years. It is native to the Indian subcontinent [3]. Tulsi (*Ocimum sanctum* L.), known as the “queen of herbs”, is a plant that grows in portions of North and Eastern Africa, China, Hainan Island, and Taiwan [4]. Tulsi is frequently referred to as an “Elixir of Life” in the Ayurvedic tradition because of its curative properties and track record of treating a wide range of common ailments (Figure 1B).

There are typically considered to be three distinct types of tulsi. *Ocimum tenuiflorum*, also known as *Ocimum sanctum* L., has two cultivars that are botanically and phytochemically distinct from one another. These cultivars are known as Rama or Sri tulsi (green leaves) and Krishna or Shyama tulsi (purplish leaves) [4,5]. *Ocimum gratissimum* is a third type of tulsi, known as Vana or wild/forest tulsi. The various types of tulsi have a vast diversity in both their morphology and their phytochemical composition, which includes secondary metabolites. However, they can be distinguished from other Ocimum species by the colour of their yellow pollen, high levels of eugenol [6], and smaller number of chromosomes [7]. Although there are different species, *Ocimum tenuiflorum* and *Ocimum gratissimum* are traditionally utilised in the same manner for the treatment of conditions that are analogous to one another. This is due to the fact that *Ocimum tenuiflorum* has six times less DNA than *Ocimum gratissimum*. In the interest of maintaining coherence, the term tulsi will be utilised to refer to either *Ocimum tenuiflorum* or *Ocimum gratissimum*.

Tulsi is used to treat a number of illnesses, including microbial, fungal, bacterial, and viral infections, cancer, arthritis, chronic fever, infertility, and eye diseases, while being considered hepatoprotective, antispasmodic, analgesic, as well as antiemetic and protecting the heart. This medicinal plant has also been demonstrated to lower blood glucose levels, making it a successful diabetic treatment [2,5,6,7,8,9].

The tulsi plant is a bushy shrub or small tree (Figure 1A) native to the tropics and subtropics. It smells and tastes completely different from anything else. It reaches a mature height of three to five feet. When making Ayurvedic remedies, tulsi leaves are commonly used. The Ocimum plant is available in numerous preparations, including Tribhuvankirti Rasa, Muktapancamrta Rasa, Manasamitra Vataka and Muktadi Mahanjana [2,5]. There are numerous preparations of Ocimum leaves, including Mahajvarankusa Rasa, Pancamrt Rasa, and Manasamitra Vataka. Ocimum is used in cold distillation, vegetable or pulse soup, refreshing beverages, Ghrit (medicinal ghee), medicinal powders, medicinal oil, Sheeta jwarantak vati (anti-malaria pills), and tulsi tea [2,5].

Tulsi has been studied for its extracts’ potential to treat a wide range of illnesses, from the common cold to heart disease to headaches to gastrointestinal problems, so its medicinal uses are not limited to inflammatory disease. Mosquitoes and flies are just two of the many insects that can be repelled by tulsi. It could possibly be used to treat malarial fever. The possible health benefits of tulsi leaves have been the subject of research [10].

A recent study by Enegide et al. reviewed the Ocimum species and its ethnomedicinal uses in 2021 [8]. There are 68 species of Ocimum, 2 of which are widely used in cooking and medicine: *Ocimum basilicum* and *Ocimum tenuiflorum*. Here, we present a few examples of Ocimum species and where they can be found:***Ocimum americanum* L**.: The leaves of the *Ocimum americanum* plant have eugenol, pinene, myrcene, camphene, sabinene, bicyclogermacrene, bisabolene, bisabolene, 1,8-cineole, limonene, fenchone, linalool oxide, linalool, borneol, camphor, 4-terpineol, and cis-piperitol [11]. *Ocimum americanum* is native to the equatorial regions of Africa, the Indian subcontinent, China, and Southeast Asia. It has since become naturalized in Queensland, Christmas Island, and other parts of tropical Australia. People have used it to treat coughs, ulcers, tuberculosis, haemorrhoids, stomach pains, and problems with the eyes and ears. It has also been used to treat stomach aches, diarrhoea, diabetes, and constipation [9,12,13,14].***Ocimum basilicum* L.**, also referred to as sweet basil, is a plant that is indigenous to the Indian subcontinent, Southeast Asia, Russia, Ukraine, Cameroon, Africa, Guinea, Mali, Mexico, Central America, South America, and a variety of islands in the ocean. It is also cultivated in a number of countries around the world. Patients who suffer from cardiovascular disease, diabetes, chronic pain, fever, vomiting, diarrhoea, and other conditions are given this as a preventative measure in addition to a treatment for their illness. In addition to having properties that make it an effective sedative, it can also be used to treat skin infections, bites from snakes, and stings from insects [15,16,17,18]. Compounds such as pinene, myrcene, 4-hexen-1-ol acetate, 4-eucalyptol, cis-linaloloxide, 1,6-octadien-3-ol,3,7-dimethyl, methyl ethyl cyclopentene, l-menthol, l-(-)-menthol, and estragole were obtained from *Ocimum basilicum* leaves through other compounds that can be found in the leaves include N-cyano-3-methylbut-2-enamine, formic acid, cyclohexyl ester, eugenol, cyclohexyl phenyl hydrazide, citral, 4-methyl-1-(1-methylethyl) cyclohexane, phenol 2,3,5-trimethyl, copaene, cis-7,10,13,16-docos [19,20].***Ocimum tenuiflorum* L. (*Ocimum sanctum*)**, also known as Holy basil or tulsi, has been naturalized in Kenya, Fiji, French Polynesia, the West Indies, and Venezuela. It is originally from China, the Indian Subcontinent, Southeast Asia, New Guinea, and Queensland, Australia. In traditional medicine, it is employed to manage and treat a variety of conditions, including headaches, fevers, coughs, the common cold, influenza, sore throats, colic pain, asthma, diarrhoea, digestive disorders, bronchitis, influenza, insomnia, arthritis, and malaria fever. In addition, it can be used as a treatment for scorpion stings and snakebites [21,22,23]. The chemical compounds eugenol, cyclohexane, 1,2,4-triethene, and caryophyllene can be found in the extracts that are made from the leaves. Compounds such as the following have been utilized: benzenemethenamine; N,N-,4-tetramethyl-; 10-heptadecen-8-ynoic acid; cyclopentane, cyclopropylidene-; Z,Z-4,16-octadecadien-1-ol acetate; benzenemethenamine; and 3′,8,8′-Trimethoxy-3-piperidyl-2,2′-binaphthalene-1,1′,4,4′-tetrone.***Ocimum gratissimum:*** The *Ocimum gratissimum* plant is indigenous to India, China, Nigeria, and both Australia and New Zealand. Diabetes, infections of the upper respiratory tract, pneumonia, epilepsy, fever, convulsions, diarrhoea, headaches, and flu are just some of the conditions that the leaves, stems, roots, and flowers of this plant are used to treat and prevent [24,25,26,27,28,29,30,31]. Compounds such as eugenol, methyl eugenol, cis-ocimene, trans-ocimene, pinene, camphor, germacrene-D, trans-caryophyllene, farnesene, l-bisabolene, thymol, methyl chavicol, linalool, limonene, and methyl eugenol can be found in the extract of the *Ocimum gratissimum* plant.***Ocimum kilimandscharicum*:** The camphor basil is a plant that is native to India, Thailand, Ethiopia, Tanzania, Kenya, Uganda, and Sudan. It is also cultivated in Uganda and Kenya. Cough, cold, measles, abdominal pain, measles, diarrhoea, and diarrhoea are some of the conditions that it is used to treat. In addition to these uses, it is also put to use as an insect repellent and for the control of pests in storage.***Ocimum campechianum*** Mill., (the Amazonian basil) can be found all over the Americas. Any portion of the plant, such as the fruits, seeds, flowers, leaves, bark, roots, and so on, can be used to produce plant-based natural constituents, meaning that any part of the plant may have active ingredients. The mixtures of secondary products found in plants are often what give plant materials their positive therapeutic properties. This idea is congruent with the idea that a particular plant’s combination of secondary metabolites has therapeutic properties that are specific to that plant’s species or group of species [32]. Tulsi has a variety of chemical components, including carvacrol, caryophyllene, elemene, eugenol, linalool, rosmarinic acid, oleanolic acid, germacrene, and ursolic acid (Table 1). Tulsi is thought to have stimulating and diuretic properties [33]. The leaves of medicinal herbs can also be used to produce volatile and fixed oils. Daily intake of its leaves and products is said to prevent ailments, increase health, lifespan, and happiness. The nutritional ingredients of tulsi are presented in Table 2. Several scientific studies have examined the potent therapeutic potential of tulsi, including numerous experiments on humans and animals. These studies have revealed that tulsi has a variety of therapeutic benefits, including anti-inflammatory, anti-pyretic, anti-allergic, anti-asthmatic, anti-tussive, anti-ulcer, anti-emetic, anti-spasmodic, mosquito repellent, anti-diarrhoeal, antioxidant, anti-stress, hepato-protective, cardio-protective, neuro-protective, anti-hypercholesterolemic, antidiabetic, anti-coagulant, adaptogenic, anti-thyroid, anti-cataract, anti-carcinogenic, radioprotective, anti-hypertensive, analgesic, diuretic, antifertility, anti-ulcer, anti-leukodermal, anti-microbial (including antiviral, antibacterial, antifungal, anti-protozoal, anti-malarial, anti-helminthic), anti-arthritic, anti-toxic, wound heal effect, immunomodulatory, CNS depressant, memory enhancement activities [34].***Ocimum Sanctum* (*Ocimum tenuiflorum* L.)**Multiple pharmacological effects of *Ocimum sanctum* have been documented and described in the literature. It has been shown by Rahman et al. [35] that *Ocimum sanctum* has antimicrobial activity against many different types of bacteria. These include *Staphylococci* sp., *E. coli*, *Shigella* sp., *S. aureus*, *Enterobacteria* sp., *P. aernginosa*, *S. Typhi*, *Staphylococci* sp., *C. albicans*, and *K. pneumonia* [35]. *Ocimum sanctum* extract was found to be a potent anti-tuberculosis agent in an in vitro study conducted by Farivar et al. [36]. Several different fungi, including *A. solani*, *C. guillermondii*, *C. capsici*, *Curvularia* sp., *F. solani*, *H. oryzae*, and *A. flavus*, were found to be effectively combated by *Ocimum sanctum*’s aqueous, hexane, chloroform, and n-butanol extracts and essential oil, as reported by Khan et al. [37]. Significant antioxidant activity was observed in vitro and in vivo when *Ocimum sanctum* was extracted using methanol, hydroalcoholic, and aqueous solvents, as reported by Kelm et al. [38]. Oral consumption of *Ocimum sanctum* significantly protects liver and aortic tissue from hypercholesterolemia-induced peroxidative damage, as reported by Geetha et al. [39]. Blood sugar levels in streptozotocin-induced and glucose-fed perglycaemic diabetic rats were significantly lowered after oral administration of *Ocimum sanctum* extract, as documented by Siva et al. Gholap and Kar found similar results, namely that *Ocimum sanctum* decreased cortisol and glucose levels in the blood and exhibited elicited antiperoxidative activity in their study [40,41]. Researchers Aruna et al. found that the incidences of squamous cell carcinoma and hematoma in experimental rats were significantly reduced when they were given *Ocimum sanctum* leaves [42]. *Ocimum sanctum* aqueous leaf extract significantly reduced hydroxyl (OH) radical-induced deoxyribose degeneration, according to research by Ganasoundari et al. [43]. Additionally, they demonstrated that WR2721 and *Ocimum sanctum*’s synergistic activity produced a more potent effect against OH radical activity than either agent alone [43]. Using an excision model, Shetty et al. tested the effects of an aqueous leaf extract from *Ocimum sanctum* on tumour necrosis factor-alpha (TNF-Alfa) in laboratory rats. The rate of epithelization and wound contraction was shown to be significantly increased by the *Ocimum sanctum* extract, indicating a significant wound healing effect. The oil of *Ocimum sanctum* demonstrated significant antiulcer activity in a study of aspirin, indomethacin, alcohol (ethanol 50%), histamine, reserpine, serotonin, and stress-induced ulcers in rats [44]. Laboratory rats’ humoral immune responses were found to be altered after exposure to a steam-distilled leave extract *of Ocimum sanctum*, as reported by Mediratta et al. [45]. This may be because of mediators released during hypersensitivity reactions, tissue responses to these mediators, or both [46]. *Ocimum sanctum*’s immunomodulatory effect was also demonstrated by Godhwani et al. in a separate experiment using widal and sheep erythrocyte agglutination tests. *Ocimum sanctum* has been shown to be effective in the treatment of asthma and related conditions, as demonstrated by Sridevi et al. [47]. *Ocimum sanctum* was reported to have the potential to stabilize mast cells, suppress IgE, and inhibit the release of inflammatory mediators, suggesting it is responsible for these effects [48]. According to research done by Ravindran et al. [49], the *Ocimum sanctum* has been found to have anti-stress properties due to the fact that it helps restore normal levels of neurotransmitters in the body after being exposed to noise stress. Researchers in the past demonstrated that the essential oil of *Ocimum sanctum* possessed powerful anti-helminthic activity by using *Caenorhabditis elegans* as a model organism. Experiments were performed on various extracts of *Ocimum sanctum* stem and leaves (n = 132) to test for anticonvulsant activity. The experiments used the maximal electroshock model as the experimental design, and phenytoin was used as the standard. It was discovered that extracts of the leaf and stem made with ethanol and chloroform produced significant preventive effects against toxic convulsions induced by trans-corneal electroshock. These convulsions were caused by toxic electric shocks delivered through the cornea. *Ocimum sanctum* has a strong cardioprotective effect against myocardial agents, as has been demonstrated in animal studies [50]. Antivenomous effects of *Ocimum sanctum* have been studied and found to be effective against venomous dog, snake, scorpion, and insect bites [50].*Ocimum sanctum*, as discovered by Sood et al., significantly shielded isoproterenol-induced myocardial necrosis in experimental rats by boosting endogenous antioxidants’ activity [51]. Furthermore, *Ocimum sanctum* L. alcoholic extract was found to improve scopolamine-induced amnesia and age-related memory loss in mice. Step-down latency (SDL) and acetylcholine esterase inhibition were both significantly enhanced by *Ocimum sanctum*. This approach might be helpful for patients suffering from dementia, Alzheimer’s disease, and other types of cognitive impairment [52]. 

## 2. Phytochemical Composition

Table 1 provides a detailed breakdown of the phytochemical make-up. Cirsimartin, cirsi-lineol, isothymusin, and isothymonin are methylated flavones derived from apigenin and luteolin that can be found in tulsi. Terpenes such as ursolic, triterpenic, andoleanolic acids, the sesquiterpene hydrocarbon caryophyllene, the oxygenated monoterpene carvacrol, caffeic, and rosmarinic acids, as well as the phenylpropenes eugenol and its methyl ether, are also present in huge quantities. Tulsi contains flavonoidsandphenolics, the majority of which are well known, such as neo-lignans, tannins, triterpenoids, sterols, cerebrosides, alkaloids, and saponin, known for their biological effects both in vivo and in vitro, including antioxidant or prooxidant, cytotoxic, antiviral, hepatoprotective, anti-inflammatory, and anticancer [16,17,18,54].

Additionally, the tulsi essential oil includes a significant amount of eugenol (>70%) recognised for its antioxidant, anti-inflammatory, cytotoxic, and antimicrobial activities [55]. On the basis of their specific molecular targets, the introduction of novel bioactive molecules with natural origins, notably from plant sources, may be thought of as a new and effective therapeutic strategy to treat various types of human cancers [56]. Additionally, oxidative stress is a significant factor in the pathogenesis of various cancer forms. Antioxidants have so received a lot of attention as a unique therapeutic approach for cancer. Studies have revealed that oxidative stress and inflammation are related phenomena that play a role in cancer. The existence of anticancer capabilities in tulsi leaves has been thoroughly established [57]. Tulsi leaves also contain a significant amount of eugenol 4200–4970 ppm. With this background, the anticancer effects of tulsi leaves and its major phytoconstituent, eugenol, to treat various cancer types are highlighted in the current review.

### 2.1. Eugenol

It has been reported that different species of Ocimum plant extracts prepared using ultrasound-assisted extraction techniques have varying total polyphenol and total flavonoid contents [58]. When compared to other Ocimum species, *O. basilicum* (246.2 mg GAE/g) and *O. gratissimum* (245.3 mg GAE/g) had significantly higher levels of geranyl acetate ethyl ester. Eugenol, a naturally occurring bioactivechemical, also known as 4-allyl-2-methoxyphenol, is a phenylpropanoid with a substituted guaiacol allyl chain. Holy basil or tulsi leaves (Lamiaceae), ginger (*Zingiberofficinale*), oregano (*Origanum vulgare*), clove (*Eugenia caryophyllata*), peppers (Solanaceae), thyme (Lamiaceae), turmeric (*Curcuma longa*), and the bark and leaves of cinnamon (*Cinnamomum verum*), have all been found to contain eugenol [26]. The two main natural sources of eugenol are cloveand cinnamon, which together account for 45–90% and 20–50%, respectively [28]. However, commercial-level extraction of eugenol is quite expensive and requires lengthy cultivation times; as more affordable alternatives, ginger, tulsi, and bay can be used in place of cinnamon and clove. The aerial portions of plants, such as the leaves, bark, and flowers, contain the majority of eugenol because these parts also contain a significant amount of essential oils [29,30].

Tulsi leaves also contain a significant amount of eugenol, often between 40 and 71%. However, the amount of eugenol in various plant sections fluctuates according to the season. According to studies, fall harvests of eugenol produce the highest yields when compared to summer types [31]. In the essential oil, the leaves, and the inflorescence, eugenol was found to make up 13.8%, 23.7%, and 7.5% of the total volatile compounds. Eugenol was found to be the most abundant chemical in tulsi leaves from all over the world, including those grown in India, Brazil, Bangladesh, Cuba, and Germany. Essential oil extracted from the *Ocimum gratissimum* plant was chemically analysed, and 67 percent eugenol was discovered to be present.

Numerous anti-inflammatory, antioxidant, antiallergic, antimutagenic, anticancer, and analgesic properties can be found in eugenol (Figure 1B) [21,31,59,60,61]. Eugenol treatment in combination with conventional medications has significant potential and ability to eradicate drug-resistant strains, forming a part of natural essential oil offers far less downsides than other types of synthetically produced substances. The majority of drug combination treatments are used to treat cancer [23]. When combined with a chemo-inhibitory medication, eugenol exhibits a synergistic action that significantly reduces drug toxicity on healthy cells [62]. A small amount of eugenol combined with gemcitabine in an in vitro trial increased the effects of the medicine without having any negative effects on healthy cells [63]. The US Food and Drug Administration (FDA) has determined that eugenol is not carcinogenic and does not cause mutations [64]. Eugenol is a guaiacol substitute with an allyl chain. Eugenol isformed from its precursor molecule phenylalanineand belongs to the group of chemical substances known as phenylpropanoids, as explained in detail in Figure 2. Eugenol’s anticancer properties are achievedina number of ways, including triggering cell death, cell cycle arrest, inhibiting migration, metastasis, and angiogenesis on number of cancers [19].

### 2.2. Caryophyllene

Another compound, β-caryophyllene, is a sesquiterpene found in 4.9%, 1.5%, and 1.2% of volatile compounds in the inflorescence, leaves, and oil, respectively, of tulsi grown in Australia. The caryophyllene is a flavour enhancer and fragrance component in many products. The caryophyllene has antibacterial properties as well [65,66].

### 2.3. Ursolic Acid (UrsA)

One of the most common and extensively researched pentacyclic triterpenes is 3-hydroxy-urs-12-en-28-oic acid, also known as UrsA, having the formula C_30_H_48_O_3_ and a molecular mass of 456.7 g/mol. UrsA is a terpene that is a secondary metabolite of plants; it is typically soluble in organic solvents but insoluble in water. Tulsi, apples, rosemary cranberries, bilberries, peppermint, oregano, and prunes are some of the foods that contain significant amounts of the UrsA compound. Urs Acid, which is used to treat ulcers, was extracted from the leaves and stems of *Ocimum forskolei* (Benth) [44]. *Ocimum sanctum* (L.) leaves induced antiproliferative and antistress therapy for rheumatoid arthritis. Additionally, UrsA decreased the level of Bcl-2 to trigger apoptosis in human MCF-7 cells [59,67].

In their study using MDA-MB-231, Yehya et al. [68] found that UrsA suppressed cancer cell invasion, migration, and proliferation, as well as the formation of cell colonies. Moreover, UrsA has been shown to significantly reduce the expression of u-PA and MMP-2 while simultaneously increasing the expression of PAI-1 and TIMP-2. The expression of u-PA, TIMP-2, PAI-1, and MT1-MMP has also been reduced as a result of UrsA. In addition, Kim et al. [69] investigated whether UrsA has the capability of inducing apoptosis in MDA-MB-231 human BC cell through both extrinsic death receptor pathways and intrinsic mitochondrial death pathways. The results of an investigation using immunoblotting demonstrated that UrsA stimulated the Fas receptor, which was then followed by caspase-8 and PARP activation. In addition to this, UrsA raises the level of expression of Bax, causes the release of cytochrome C, lowers the level of Bcl-2, and activates capsases-9. Additionally, UA decreased the level of Bcl-2 to trigger apoptosis in human MCF-7 cells [70].

A key role in cancer prevention is played by UI which has been shown to decrease phosphorylation of Akt and the PI3K/Akt/mTOR signalling pathways [71]. In addition, UrsA has been shown to prevent the colony formation of WA4 mammary cells in vivo. In addition, UrsA has the potential to induce apoptosis, arresting the WA4 cell cycle in the G1 phase by increasing levels of cleaved caspase 3. UrsA has been shown to be a promising phytochemical, and its use in the pharmaceutical industry for the treatment of breast cancer (BC) has been the consensus of numerous research studies. The scholarly community is turning to dietary phytochemicals as a source for new drugs because of their low risk profile. The effectiveness of numerous phytocompounds as anti-BC agents has been widely acknowledged. Recent studies have shown promise for UrsA as a treatment option for carcinoma breast. In both in vitro and in vivo models, UrsA has been found to exhibit potent anti-breast cancer potential by inducing apoptosis, arresting the cell cycle, and preventing proliferation, angiogenesis, and metastasis [72]. More research is necessary to determine whether UrsA can sensitize and enhance the chemotherapeutic drugs. To improve their bioavailability in clinical trials, UrsA requires further study. As a result, creating synthetic analogues is the most effective method for addressing this issue. Combinatorial therapy is another option for addressing the issue of bioavailability. Moreover, there is a need to investigate novel targeted proteins, signalling cascades, epidemiological studies, clinical trials, and safety profiles. For the most part, UrsA has been shown to be a promising phytocompound for the treatment of BC in a variety of in vitro and in vivo studies.

### 2.4. Rosmarinic Acid (RA)

Rosmarinic acid, also known as RA, is a type of flavonoid that is frequently discovered in plants belonging to the Lamiaceae family. Tea, herbs, cooking condiments, spices, and fruits all make use of RA-rich plants such as *Perilla frutescens* (L.), Britton, Rosmarinus officinalis L., and Melissa officinalis L. These plants are popular all over the world and are used in a variety of applications. Because of its nutritional qualities and the fact that it has been demonstrated to possess powerful antioxidant activity [73,74], RA is used to make people healthier. RA is an ester of caffeic acid and 3,4-dihydroxyphenyllactic acid, andis one of the primary phenolic compounds found in *O. sanctum* [75]. The leaves of *O. sanctum* were extracted with ethanol (EEOS) and analysed using a trusted LC-MS technique by Shanmugam et al. in 2012. They then identified RA and UrsA as the functional molecules in EEOS. It has been found that rosmarinic acid has powerful antioxidant properties. It protects cells from free radicals, which would otherwise destroy them. Furthermore, cellular damage is brought about by an excess of oxidation in the body. When this acid is present, it inhibits oxidation from happening in excess. In addition to its antioxidant properties, RA has anti-inflammatory properties. Pegenin is another compound in the mixture that can perform the same task. In addition to these two components, eugenol is tulsi’s primary anti-inflammatory catalyst. It is the primary factor that helps keep blood sugar levels stable. It increases insulin secretion by stimulating pancreatic beta cells. It has been found that RA has antimicrobial, immunomodulatory, diabetic, anti-allergic, anti-inflammatory, hepato-protective, and renal-protective properties [76]. In addition, the utilization of RA has a prospective application in the management and prevention of cancer [77]. Research on RA is currently being conducted to investigate its potential uses in the treatment and prevention of cancer [78].

When the leaves of the *Ocimum tenuiflorum* L. plant are soaked in 95% ethanol for two weeks, filtered and then dried, 7.86 mg/g of rosmarinic acid is obtained. This acid inhibits the invasion of head and neck squamous cell carcinoma cells [79]. The dry leaves of *Ocimum basilicum* L., extracted in 99% methanol, had RA 3.01 mg/g, which demonstrated antiproliferative activities against cervical, T-cell, and breast carcinoma [80]. Ovarian carcinoma cells were made more sensitive to cisplatin (DDP) by an ethanol extract of dried rosemary leaves [81]. Anti-inflammatory targets of RA for cancer therapy include cyclooxygenase-2 (COX-2) and nuclear factor-kB (NF-kB). In lung, breast, and liver cancer cells, RA was found to inhibit COX-2 activity and negatively regulate ERK1/2, resulting in anti-inflammatory effects [82]. Research showed that RA controlled EMT-related protein expression and reduced tumour cell invasion [83].

RA promoted EMT by upregulating E-cadherin, inhibiting N-cadherin, and concomitantly inhibiting matrix metalloproteinases (MMPs). This resulted in osteosarcoma, pancreatic cancer, and colorectal cancer having a reduced ability to invade [84,85,86]. Further study discussed that RA was able to inhibit EMT, increase chemosensitivity, and suppress the expression of zinc finger E-box binding homeobox 1, snail1, and twist1 [68].

### 2.5. Apigenin

Apigenin (APG) is an edible flavonoid that is also known as 4,5,7-trihydroxyflavone. In recent years, it has gained popularity as a health-promoting drug due to its low intrinsic cytotoxicity and differential effects on normal versus cancer cells. These two factors contribute to the drug’s ability to target cancer cells more specifically than normal cells. This is specially the case when compared, in particular, with other polyphenols that are structurally related polyphenols [87]. Apigenin is a polyphenol that can be found in a variety of flavonoids. Apigenin’s potent antioxidant and anti-inflammatory activities are a significant contributor to its possible cancer-preventive implications [88]. Apigenin also inhibits the growth of cancer cells. Quite remarkably, apigenin significantly engages in the prevention of cancer by meaningfully contributing to the induction of apoptosis in various cell lines as well as animal models [89]. Apigenin has been shown to have this effect in both animal models and cell lines. Apigenin is useful as a source of medications due to its low toxicity and apparent role in reducing cancer treatment resistance. Apigenin’s antioxidant properties show that it plays a significant role in the regulation of free radical formation, modulation of oxidative stress and inflammation, and management of cancer. This flavonoid appears to exhibitanticancer activities by regulating multiple cellular signalling pathways, some of which may involve angiogenesis, apoptosis, the cell cycle, and other genetic pathways.

Apigenin strongly suppressed colorectal cancer cell growth, proliferation, migration, invasion, and organoid development by inhibiting the Wnt/-catenin signalling pathway [90]. Combination therapy with apigenin and cetuximab also decreases the expression of p-epidermal growth factor receptor, p-Akt, p-signal transducer and activator of transcription 3, and cyclin D1 [91]. Through the signal transducer and activator of the transcription (STAT-3) pathway, apigenin suppressed the expression of MMP-2, MMP-9, and vascular endothelial growth factor (VEGF), all of which play roles in cell migration and invasion. Apigenin effectively blocked STAT3 transcriptional activity, decreased STAT3 nuclear localization, and inhibited STAT3 phosphorylation [92]. Additionally, STAT3 transcriptional activity, STAT3 phosphorylation, and STAT3 nuclear localization were all effectively suppressed by apigenin [93]. Apigenin suppressed ERK1/2 and P90RSK phosphorylation while activating AKT and S6 phosphorylation [94]. Two kinases, AKT and ERK, were both inhibited by apigenin. Moreover, apigenin boosted the antitumour activity of ABT-263 in colon cancer cells by decreasing the expression of pro-survival regulators AKT, Mcl-1, and ERK [95].

It was found that increased p53 accumulation was linked to increased Bax detection. Previously, it was believed that apigenin increased cisplatin’s cytotoxicity by activating p53 accumulation and p53-regulated proapoptotic gene expression [96]. Apigenin is a useful tool for controlling mucin production and gene expression because it modulates the nuclear factor-kappaB signalling pathway in airway epithelial cells.

Apigenin inhibits the production of the chemokines interleukin-6 and interleukin-1, which are released in response to tumour necrosis factor alpha [97]. Apigenin increases thymidylate synthase expression, which makes colorectal cancer cells more vulnerable to apoptosis induced by 5-fluorouracil [98]. Apigenin induces TRAIL-induced anti-tumour activity in lung cancer cells by blocking the pro-survival regulators ERK, nuclear factor kappaB (NFkB), and AKT. Apigenin, in contrast to an IKK inhibitor, directly binds with IKK, inhibits IKK kinase activity, and dampens NFkB/p65 activation in cancer cells [99]. Apigenin inhibited the growth of transgenic prostate adeno-carcinoma by blocking FoxO/PI3K/Akt signalling [100]. The PI3K/Akt/mTOR pathway is involved in the induction of apoptosis and autophagy, and apigenin blocks this pathway [101]. During the G2-to M transition, apigenin inhibited the expression of cyclin B, cyclin A, and cyclin-dependent kinase-1, all of which are involved in controlling the cell cycle [102].

When apigenin is applied to cancer cells, DNA damage, apoptosis, and cell cycle arrest in the G2/M phase are induced, which ultimately leads to a reduction in cancer cell proliferation in vitro and in vivo [103]. A recent study found that apigenin inhibits the mRNA expression of vascular endothelial growth factor in response to hyperoxia [104]. Apigenin inhibits angiogenesis by decreasing HIF-1alpha and vascular endothelial growth factor expression in tumour tissues [105]. As DNA fragmentation following PARP cleavage showed, apoptosis was induced by apigenin treatment. These results were linked to an elevated Bax/Bcl-2 ratio, which indicates a shift favouring apoptosis [106]. When exposed to apigenin, Bcl-xL and Bcl-2 levels were found to decrease, while Bax protein levels were found to rise [100].

### 2.6. Carvacrol

Significant research has been done on the natural compounds carvacrol (5-isopropyl-2-methylphenol) and its isomer thymol (2-isopropyl-5-methylphenol). Both of these compounds have multiple biological effects. Both are monoterpenoid phenols, which are the primary compounds that can be found in the essential oils of many plants belonging to the Lamiaceae and Verbenaceae families, such as oregano (*Origanum vulgare* L.), thyme (*Thymus vulgaris* L.), and “ale-crim-da-chapada” (Lippiagracilis) [57,107,108]. These compounds exhibit anti-inflammatory.

Inhibition ofthe viability and proliferation of lung cancer cells (A549 cell line) and induction ofearly apoptotic characteristicswere also achieved by 1 mM carvacrol and 0.5 mM carvacrol [87,109,110]. Tyrosine kinase receptor (AXL) expression was suppressed and malondialdehyde (MDA) and 8-hydroxy-2′-deoxyguanosine (8-OHdG) levels were raised, primarily responsible for these effects [109,110]. Carvacrol exhibited anticancer effects in relation to hepatocarcinomas (HepG2 cell line), inducing cell death and antiproliferative effects in a concentration-dependent manner [14,111]. The mitochondria-mediated pathway was responsible for the downregulation of cell proliferation and the induction of apoptosis, which was accompanied by the activation of caspase-3 and the downregulation of Bcl-2. Apoptosis may also be triggered by the mitogen-activated protein kinases (p38) and extracellular signal-regulated kinases (ERK) protein [112]. Similarly, Ref. [14] showed that carvacrol at a concentration of 650 M slowed cell cycle/mitosis and induced cell death within 24 h of incubation, with an accumulation of cells in the G1 phase and a reduction in cells in the S phase. Additionally, incubation with carvacrol (115 M) decreased cell viability and significantly increased the number of early apoptotic cells in the colorectal cancer (Caco-2 cell line) [113]. Proliferation of HCT116, LoVo, and HT-29 cells was also suppressed [114]. Cell cycle arrest at the G2/M phase was another effect of carvacrol, which also decreased Bcl-2, metalloproteinases 2 and 9 (MMP-2 and MMP-9), phosphorylated extracellular signal-regulated kinases (p-ERK, p-Akt), and cyclin B1 and increased phosphorylated jun N-terminal kinase (p-JNK), and Bax [115]. Carvacrol has been shown to reduce the viability of breast cancer cell lines MDA-MB231 and MCF-7 [67,116].

It has also been proposed that the pharmacological mechanism of action is related to the TRPM7 pathway [117]. When compared to thymol, thujone, 4-terpineol, 1,8-cineol, bornyl acetate, and camphor, the cytotoxic effects of carvacrol were found to be significantly stronger [116]. Recently, scientists also reported an increase in the levels of the protein Bax, as well as the cleavage of the enzyme poly [ADP-ribose] polymerase 1 (PARP-1) and caspase-3, which are all indicators of an apoptotic effect [118].

Carvacrol inhibited cell growth by 1.2 times more in MDA-MB231 cells than in U87 cells [119]. Carvacrol treatment suppressed MDA-MB 231 cell proliferation and was accompanied by apoptosis induction characterized by elevated levels of Bax, decreased mitochondrial membrane potential, cytochrome C release, caspase activation, PARP cleavage, an increase in the sub-phase G0/G1 of the cell cycle, and a decrease in the number of cells in the S phase [120].

Reduced expressions of cyclin D, cyclin E, phosphoinositide-3-kinase (PI3K), phosphoinositide-3-kinase (p-AKT), retinoblastoma protein (pRB), and phosphoinositide-3-kinase (p-AKT) led to an increase in cells in the G0/G1 phase and a decrease in cells in the S and G2 phases [70].

## 3. Anti-Inflammatory Potential

Regularly, people with inflammatory diseases utilise glucocorticoids or nonsteroidal anti-inflammatory medicines (NSAIDs). However, these medications have been associated with major adverse effects include bleeding and stomach ulcers. These medications also showed limited remedial efficacy, which commonly leads to patient treatment discontinuation [121,122]. As a result, the pharmaceutical industry has concentrated its efforts on finding new bioactive chemicals [122].

The development of novel medications that target various illnesses, such as those linked to inflammatory responses, which are frequently linked to oxidative stress, is attracted by the bioactive components from medicinal plants [123]. Most of these substances have the ability to reduce oxidative stress and inflammatory reactions. Additionally, these substances may contribute to a preventative strategy for enhancing life quality through consumption of a diet high in them [124].

Eugenol’s anti-inflammatory properties result from the suppression of production of prostaglandins and chemotaxis of neutrophils and macrophages, as represented in Figure 3A. Additionally, in vitro research shows that this bioactive substance blocked nuclear tumour necrosis factor (TNF)-induced factor-B (NF-κB) activation, suppressed leukotriene C4 and 5-lipoxygenase (5-LOX), and stopped LPS-stimulated cyclooxygenase (COX-2) activity macrophages [125]. The expression of COX-2 is induced by growth factors, LPS and cytokines [126]. Eugenol displayed a decrease in inflammation of reducing TNF-α and neutrophil infiltration during pulmonary disease in animals. In animal models of pulmonary disease, eugenol was shown to have anti-inflammatory effects, as measured by a reduction in both TNF- and neutrophil infiltration. When the substance was administered at a dose of 160 mg/kg body weight in lung polymorphonuclear (PMN) infiltration [127], there was a decrease in the alveolar collapse that was observed. In addition, eugenol prevented chemically induced dysfunction in macrophage cells and restored balance to pro/mouse peritoneal macrophages, which are cells that produce anti-inflammatory mediators [127,128,129].

The process of leukocyte recruitment to tissue is crucial to inflammation. Eugenol has been shown to reduce leukocyte rolling, adhesion, and migration to the inflammatory site in this regard [130]. These findings are corroborated by additional research on mice treated with lipopolysaccharide (LPS) in which eugenol decreased lung neutrophil/macrophage infiltration and attenuated the production of inflammatory cytokines (IL-1, IL-6, and TNF-α), as well as the activation of NF-κB [131]. In an animal model of lung injury brought on by diesel exhaust particles, eugenol thereby lowers the inflammatory response. By preventing PMN infiltration and apoptosis by caspase-3 cleavage, eugenol treatment reduced pulmonary inflammation; however, the results in combating oxidative stress were not great. The pulmonary mechanics and airspace collapse, which are affected by diesel particlesand measured by pneumotachograph, improved as a result [132]. These findings showed that eugenol has the potential to be used as a treatment for the negative consequences of exposure to air pollutants through mechanisms partially mediated by its anti-inflammatory properties.

Additionally, in an in vivo study of ovalbumin-instigated allergic asthma, eugenol reduced ovalbumin-specific IgE levels, as well as interleukin-4 and -5 (IL-4 and-5), the primary cytokines in various types ofallergic reactions, preventing the emergence of a Th2-type immune response [133,134]. This signifies the claim that eugenol’s anti-asthmatic impact, which led to a decrease in airway resistance (AWR), was mostly dependent on the inflammatory response’s lowering [135]. According to this information, eugenol may be used therapeutically and strategically in asthma patients.

In a study by El Motteleb and his colleagues [136], using an ischemia and reperfusion (I/R) model, the effectiveness of eugenol in preventing liver damage was examined. Eugenol reduced TNF-α and NF-κB expression, reduced myeloperoxidase (MPO) activity, and reduced inflammation. It also altered an oxidative marker. Malondialdehyde (MDA) was decreased, and GSH levels were raised. The liver structural and functional damage was ameliorated as a result of eugenol’s powerful impact [137,138,139]. The lowering of inflammatory mediators and the regulation of the redox state caused by eugenol reduced liver damage, suggesting a potential use against hepatic I/R injury.

Eugenol, therefore, has the potential to act as an anti-inflammatory substance (Table 3), enabling it to replace certain NSAIDs in a variety of diseases. Additionally, it could be applied to the creation of novel, wide-ranging drugs to treat diseases such ascancer and others that are associated with inflammatory processes.

## 4. Antioxidant Potential

Oxidative stress is a condition where the body’s defensive and offensive systems are out of balance. It is caused by an excessive production of reactive oxygen species (ROS), such as superoxide radicals, hydroxyl radicals, and hydrogen peroxide, which the body’s antioxidant mechanisms are unable to neutralize [159]. This process has harmful effects and alters the usual redox state, which is linked to lipid peroxidation and cellular damage [160].

Due to an excess of free radicals, many human-related ailments, such as cancer, liver disease, diabetes, renal disease, arthritis, Parkinson’s disease, cardiovascular disease, Alzheimer’s, and AIDS problems, are triggered and exacerbated [161,162]. According to reports, fruits and vegetables with a high concentration of phytonutrients, such as polyphenols, anthocyanin, and flavonoids, are effective at scavenging free radicals [163]. Eugenol playsan important role as an antioxidant. (Table 3)

Eugenol is a powerful antioxidant that blocks the activity of monoamine oxidase. It has been demonstrated to possess neuroprotective properties [63]. It is known to filter free radicals, prevent the production of reactive oxygen species (ROS),boost cyto-antioxidant capacity, to safeguard proteins from damage, protect microbial DNA from oxidative damage (Figure 3B). Eugenol can also repair oxidative damage, inhibit cancer-causing mutations, and eliminate damaged molecules [164,165,166]. The antioxidant properties of eugenol can be inferred through its mechanism of reducing free radicals by the provided H-atoms [167].

The impact of the free radical scavenger 2,2-diphenyl-1-picrylhydrazyl(DPPH) is explained by certain elements’ propensity to provide hydrogen, particularly those with phenolic groups in their structures [168]. Eugenol exhibits strong antioxidant properties. It shows strong anti-DPPH radical effect having a half maximal inhibitory concentration (IC_50_) of 11.7 µg/mL and prevents the production of ROS in human neutrophils activated by phorbol 12-myristate 13-acetate (PMA) IC_50_ at 1.6 µg/mL. Additionally, eugenol shows enhanced myeloperoxidase (MPO) hindrance in human leucocytes with an IC_50_ of 19.2 µg/mL and inhibits the production of nitric oxide having an IC_50_ of 19.0 µg/mL [169]. When the concentration of eugenol was decreased from 1.0 to 0.1 µM/mL, it was able to eliminate over a 4/5 part of the DPPH radicals and lower the power of radicals [170,171].

Studies have revealed that oxidative stress and inflammation are related phenomena that play a role in diseases such as cancer [172]. In this way, increased ROS production occurs in the damaged inflammatory tissue during inflammatory events, which can stimulate and play a crucial role in the signalling pathway for the production of inflammatory mediators such asproinflammatory cytokines and chemokines, leading to inflammatory cell migration [173].

## 5. Anticancer Potential

Chemoprevention drugs are essentially substances that stop the growth of cancer [174]. Their anti-carcinogenic effects are attributed to interfering with the carcinogen’s interaction with cellular DNA, changing intracellular signalling pathways as a result of preventing the progression of an initiated cell through pre-neoplastic changes into a malignant cell, hindering angiogenesis, causing cell cycle arrest, and inducing apoptosis [175,176,177]. It is thought that the apoptosis induced by chemoprevention medicines may not only limit the carcinogenesis induced by carcinogens, but also suppress the growth of tumour and improve the cytotoxic effects of antitumour drugs on tumour, which plays a key role in the antitumour therapies [178].

Tulsi has been the subject of extensive research in recent years to better understand its potential pharmacological effects. For example, an ethanolic extract of *Ocimum sanctum* (EEOS) caused Lewis lung carcinoma to undergo apoptosis [142], while an aqueous extract of *Ocimum gratissimum* prevented the spread of breast cancer by inhibiting matrix metalloproteases [179]. Additionally, it has been demonstrated that EEOS significantly affects the carcinogen-metabolizing enzymes cytochrome P450, cytochrome b5, and aryl hydrocarbon hydroxylase [180,181]. In addition, it has been shown that tulsi made as a fresh leaf paste, aqueous extract, and ethanolic extract decreases the occurrence of papilloma and squamous cell carcinoma in hamsters treated with carcinogens [182,183]. Furthermore, human colorectal cancer cells showed tulsi essential oil to promote cytotoxic and apoptotic activities [184]. The aqueous leaf extract and seed oil are said to exhibit chemo-preventive and antiproliferative activity in Hela cells [185]. Tulsi leaves are said to contain 0.7% volatile oil, which is composed of roughly 71% eugenol and 20% methyl eugenol [185]. The anticancer effect of eugenol against various cancer types is shown in Figure 4 and Table 3.

## 6. Skin Cancer

DMBA (7,12-dimethylbenz[a]anthracene) is an immunosuppressant and a potent laboratory carcinogen that targets specific organs. The chemical DMBA plays a role in the initiation of tumours. In some animal models of two-stage carcinogenesis, treatments with 12-O-tetradecanoylphorbol-13-acetate (TPA) can stimulate the growth of tumours. Studies on phytochemicals have revealed that eugenol is helpful in reducing the dermatoxic effects of xenobiotics and UV radiation. Among these, eugenol has received the most attention; research by numerous researchers has demonstrated that eugenol is helpful in preventing mouse skin carcinogenesis that is brought on by 7,12-dimethylbenz[a]anthracene (DMBA), DMBA and croton oil promotion [186,187], and DMBA initiation and TPA promotion [188,189]. It has been demonstrated that eugenol mediates the protective effects by preventing the production of superoxide and lipid peroxidation [186], reducing oxidative stress [190], inflammation [191], cell proliferation [139], and triggering apoptosis [158]. Eugenol treatment reduced the size of the tumour but had no effect on the formation of the tumour in a mouse skin cancer model [158,189,192]. The lowering of various inflammatory indicators, including nitric oxide synthase (iNOS), cyclooxygenase-2 (COX-2), prostaglandin E2(PGE2), cytokine levels (IL-6), and tumour necrosis factor-alpha (TNF-α), demonstrated that eugenol’s anticarcinogenic impact was accompanied with anti-inflammatory capabilities [123]. In an in vivo study of a skin cancer model, eugenol also showed chemo-preventive effects that reduced the frequency and size of skin tumours and improved animal survival rates by promoting inhibiting cellular proliferation, apoptosis, and limiting skin tumourigenesis at the dysplastic phase. According to molecular investigations, pre-treatment with eugenol suppressed NF-κB, decreased ornithine decarboxylase (ODC) activity, increased p53, and Bcl2 expression [193,194]. Studies utilising patient-derived primary melanoma cells demonstrated that eugenol induced a concentration-dependent suppression of cell proliferation[195]. Eugenol’s tumour-suppressive actions have been linked to the prevention of metastasis, reduced tumour size, and delayed tumour growthin human melanoma. Eugenol’s ability to prevent melanocyte invasion was studied by Pisano and colleagues [196]. By stopping the cell cycle, eugenol causes apoptosis. Additionally, Miyazawa et al. looked into how eugenol affected a B16 xenograft model [197]. Eugenol works by producing ROS, which inhibits DNA synthesis and delays the progression of cancer. Eugenol activity led to a 40% reduction in tumour growth [29]. Tulsi leaf extracts (aqueous and dry) showed significant cytotoxicity against the leukemic cell lines in an in vitro study.

## 7. Lung Cancer

The evidence suggests that eugenol may function in cancer cells as a pro-oxidant, disrupting signalling pathways, and as an antioxidant, preventing mutation. Prostaglandin synthesis is downregulated, NF-κB stimulation is inhibited, cell cycle S-phase activation is initiated, and apoptosis is induced by lowering inflammatory cytokine levels, according to theories about the molecular process [198]. Eugenol has been found to have chemotherapeutic effects against human lung cancer [140].

In an in vitro and in vivo study by Cui and colleagues to investigate the potential therapeutic effect of eugenolagainst the up-regulated level of Tripartite motif 59 (TRIM59) and p65 in NSCLC, eugenoltherapy significantly decreased xenograft tumour growth and extended the total survival of tumour-bearing mice. Eugenol inhibited p65 expression in a mechanistically sound manner, which subsequently reduced TRIM59 expression. The anti-tumoural effect induced by eugenol was completely recapitulated by TRIM59 loss. Eugenol’s tumour-suppressive activity was totally eliminated by ectopic TRIM59 expression, highlighting TRIM59′s predominate function in modulating the signalling that follows eugenol therapy. Through the NF-κB-TRIM59 route of regulation, eugenol suppressed NSCLC [199].

According to an in vitro study, a lower dose of eugenolinhibited metastasis and lung cancer cell sustainability onhuman embryonic lung fibroblast MRC-5 and lung cancer adenocarcinoma cells A549A.This was accomplished through inhibition of the PI3K/Akt pathway and prevention of MMP (matrix metalloproteinase) action. Eugenol had fatal effects on both healthy andlung cancer cells at higher concentrations (1000 µM). One of the leading causes of death for men worldwide is now lung cancer. Additionally, it is alarmingly rising among females. The purpose of Choudhury et al.’s research was to use eugenol’s anticancer characteristics to stop lung carcinogenesis caused by N-nitrosodiethylamine (NDEA) [200]. Eugenol was employed in a crucially effective manner to identify the chemopreventive mechanism underlying the NDEA-triggered mouse lung carcinogenesis model, and it was validated in the A549 human lung cancer cell line. By examining the β-catenin that serves as its master controller, Choudhury and colleagues [200] investigated the drug-resistant and most powerful cancer cells known as cancer stem cells (CSCs). It was found that the safe dose of eugenol caused apoptosis while at the same time inhibiting cell invasion in the lung tissue of mice treated with carcinogens but had no effect on healthy animals. Eugenol demonstrated remarkable chemoprotective ability in this lung carcinoma by combining cellular apoptosis and growth. As a chemoprotective drug, it strongly restrained lung cancer in the mild dysplastic stage. The molecular analysis clearly demonstrated the restriction on β-catenin nuclear transportation. After eugenol treatment, the reduced β-catenin pool and activated N-terminal Ser37 phosphorylation form led to the cytoplasmic breakdown of the protein [201]. This resulted in a significant decrease in the expression of CSCs markers such as Oct4, CD44, EpCAM, and Notcht1, whose expression is dependent on β-catenin, as demonstrated by immunocytochemistry (ICC), immonohistochemistry (IHC), and western blot (WB) studies both in vivo and in vitro. The in vitro secondary sphere-forming assay further confirmed the virulence and dramatically decreased CSCs population. Additionally, Western blot analysis showed that eugenol significantly increased β-catenin breakdown after exposure to the CK1 inhibitor D4476 in vitro. The N-terminal Ser45 phosphorylation of β-catenin, which is labelled by CK1 in the Wnt/β-catenin pathway, eventually frees up a site for crucial phosphorylation by GSK3 at the Ser37 residue to take place [200].

In A549 cells, EEOS caused cytotoxicity, increased the sub-G1 population, and caused apoptotic bodies to form. Furthermore, EEOS activated caspase-9 and -3 proteins while also cleaving poly(ADP-ribose)polymerase (PARP), releasing cytochrome C into the cytosol. Additionally, EEOS decreased the phosphorylation of Akt and extracellular signal-regulated kinase (ERK) and enhanced the ratio of pro-apoptotic protein Bax/antiapoptotic protein Bcl-2 in A549 cancer cells. Additionally, it was discovered that EEOS can, in a dose-dependent manner, inhibit the growth of LLC transplanted onto C57BL/6 mice. Overall, these findings show that EEOS promotes apoptosis in A549 cells through a mitochondrial caspase-dependent route and inhibits LLC’s in vivo development, pointing to EEOS’ potential use as a chemopreventive for lung cancer [142]. Eugenol treatment was shown to reduce MMP-2 (along with phosphate-Akt) expression in a human lung cancer cell line, inhibiting cell viability and impairing cell motility and invasion [140,143,144].

## 8. Breast Cancer

Mammary epithelial cells in females are managed by maintaining a delicate balance between the cell cycle and apoptosis. When this is disturbed, more mammary epithelial cells are produced, ultimately resulting in breast cancer [202]. There is an urgent need for novel anticancer medications with improved efficacy and selectivity due to the fatal and adverse consequences of currently used chemotherapeutic treatments.

A recent study found that eugenol has synergistic chemotherapeuticeffects (eugenol was administered with cisplatin) and makes triple negative breast cancer cells (TNBC) more sensitive to cytotoxicity. Further, the levels of IL-6 and IL-8 went down because the NF-B signaling pathway was turned off. This stopped the p50 and p65 subunits from being phosphorylated and moved them to the nucleus [203].

Tulsi has been studied for its anticancer activities against breast cancer. Eugenol’s antitumour properties in human breast cancer cells were examined by Vidhya et al. (MCF-7) [145]. A possible pharmacological candidate for triggering apoptosis in human breast cancer cells is the epoxide derivative of eugenol [46]. Dose-dependent inhibition of cell proliferation by eugenol had been reported in breast cancer [204]. Induction of apoptosis was reported by Abdullah et al. in HER2 positive (SK-BR-3) and triple-negative (MDA-MB-231) breast cancer cell lines via PI3K/AKT/FOXO3a pathway inhibition treated with eugenol in a dose-dependent manner that led to the significant increase in the protein levels of AKT serine/threonine kinase 1 (AKT), forkhead box O3 (FOXO3a), cyclin-dependent kinase inhibitor (p27), caspase-3 and -9, and cyclin dependent kinase inhibitor 1A (p21). In HER2-positive and triple-negative breast cancer cells treated with eugenol, MMP2 and MMP9 expression wasmarkedly reduced, but TIMP1 expression was significantly increased [58]. Eugenol was also found to induce autophagic cell death by increasing the levels of microtubule-associated protein 1 light chain 3 (LC3) and decrease the levels of nucleoporin 62(NU p62), both of which are indicators of autophagy [204].

Eugenol shows a particular toxicity against certain breast cancer cells at low doses, as low as 2 µM. Independent of the state of p53 and ER, this killing impact was primarily achieved by activating the internal apoptotic pathway and significantly suppressing E2F1 and its downstream anti-apoptosis target surviving [205]. It is significant to note that similar anti-proliferative and proapoptotic effects were also seen in vivo in human breast tumour xenografts. In light of the fact that eugenol demonstrates a concentration of anti-breast cancer properties both in vitro and in vivo, it is possible to use it to intensify the adjuvant treatment of breast cancer by targeting the E2F1/survivin pathway, particularly for the less responsive triple-negative subtype of the disease. Other breast cancer-related oncogenes, including NF-κB and cyclin D1, were also suppressed by eugenol. Additionally, eugenol reduced the growth of breast cancer cells in a p53-independent way and elevated the adaptable cyclin-dependent kinase inhibitor p21WAF1 protein [25]. Based on these studies, one can infer that eugenol can adversely affect the survival of various breast cancer types by controlling a wide range of intricately interrelated pathways.

## 9. Gastric Cancer

Using Wistar rats, Manikandan and colleagues examined the chemopreventive effects of eugenol on stomach cancer caused by N-methyl-N′-nitro-N-nitrosoguanidine(MNNG). When rats were given MNNG, it caused stomach cancer by preventing apoptosis and promoting pro-invasiveness and angiogenesis [50]. When eugenol was given to rats that had been given MNNG, the expression of phosphorylated IκBα (pIκBα), NF-κB (p50 and p65), and IκB kinase β (IKKβ) was greatly reduced, while significantly increasing the expression of kappaB alpha inhibitor (IκBα) [150]. Eugenol modulatory affects the NF-κB target genes and significantly downregulates PCNA, cyclin B, and cyclin D expression, while upregulatingpGadd45, p21, and p53 expression. Eugenol therapy reduced tumour incidence in a rat model of chemically generated stomach cancer to 16.66%. Through the modulation of Bcl-2 proteins, apoptotic protease activating factor 1 (Apaf-1), caspases, and cytochrome c, eugenol treatment induced apoptosis via the mitochondrial pathway and inhibited angiogenesis by altering the activity of matrix metalloproteinases (MMP), vascular endothelial factor (VEGF), and tissue inhibitor of metalloproteinase-2 (TIM-2) [206]. In a similar manner, eugenol treatment was shown to reduce MMP-2 (along with phosphate-Akt) expression in a human lung cancer cell line, inhibiting cell viability and impairing cell motility and infiltration [52,140]. Eugenol can significantly alter the NF-κB signalling pathway, which can have an impact on cancer chemoprevention and treatment [50].

## 10. Osteosarcoma

Eugenol’s molecular mechanism of action against osteosarcoma cells was investigated by Shin and colleagues. They demonstrated that eugenol can suppress the growth of human osteosarcoma cells in a time anddose-dependent manner. After 24 h of treatment with eugenol, survival rates ranged from 91.7% to 8.4% at doses of 0.5 to 10.0 mM. Eugenol decreased the viability from 84.8% to 5% as compared to the control at 2 mM concentration during the course of 8–48 h of treatment. Western blot analysis revealed elevated caspase-3, p53 and cleaved PARP levels. Additionally, it was observed that eugenol administration in osteosarcoma cells caused the breakdown of lamin A and decreased DFF-45 in the cytoplasm [207].

Eugenol has the capacity to enhance Bax, cytochrome C and triggering the caspase mechanism necessary for apoptosis. Eugenol activates caspase-3 in human osteosarcoma cells to cause apoptosis. Caspase-3 or caspase-7-deficient cells, however, displayed delayed mitochondrial processes connected to intrinsic apoptosis, indicating antitumour action [208]. Eugenol may be a promising anticancer agent for osteosarcoma cells, as proposed by Shin et al., who found that it induces death via a caspase-dependent route in HOS cells [207].

## 11. Colorectal Cancer

Eugenol had an antiproliferative effect on colon cancer cells in a dose-dependent manner. Additionally, eugenol caused a time-dependent rise in the number of cells accumulating in the sub-G1 phase, a sign of apoptosis [209]. By reducing non-protein thiols, eugenol increased the production of ROS while reducing MMP and transmitting the apoptotic signal. The DNA fragmentation that characterises apoptosis was brought on by increased ROS production in the eugenol-treated colon cancer cells. Additionally, p53 activation and PARP cleavage increased in tandem with ROS production [210]. Finally, colon cancer cells under the influence of caspase-3 underwent increased apoptosis. Eugenol was shown to be a likely candidate for inducing apoptosis in human colon cancer cells and strengthening its reputation as a potential chemopreventive agent against colon cancer [140]. In their research on NCM-460 cells (epithelial colon) and their anticancer activity against colorectal cancer cell lines, Petrocelli and colleagues used the essential oils of eugenol [53]. Eugenol was identified by Petrocelli and colleagues as a component with distinct anticancer action that targeted the altered colonic cells [53]. Eugenol at a concentration of 800 µM induced cell cycle retardation, cell death, and necrosis in Caco-2 and SW-620 cells after 72 h of treatment, but not in NCM-460 cells [158]. Eugenol significantly inhibited PGE-2, with an IC50 value of 0.37 µM, in addition to suppressing COX-2 genes in the HT-29 human colon cancer cell line. They also showed that eugenol prevented the growth of HT-29 cells and COX-2 gene mRNA expression [159].

## 12. Conclusions

Tulsi is referred to as the “Elixir of Life” based on various nutraceutical and pharmaceutical activities of tulsi as a whole or its parts, speciallyitsleaves, against a wide range of diseases. The two main natural sources of eugenol are clove and cinnamon, but commercial level extraction of eugenol is quite expensive and requires lengthy cultivation times; hence, tulsi can be an affordable alternative that would be used in place of cinnamon and clove. This review highlights the antioxidant; anti-inflammatory properties of tulsi and its leaves’ major component, eugenol, in conjugation to justify the anticancer properties. Eugenol’s anticancer properties are achieved in a number of ways, including triggering cell death, cell cycle arrest, inhibiting migration, metastasis, and angiogenesis in a number of cancers. The specific molecular targets-based introduction of novel bioactive molecules with natural origins (plant sources) combined with nanotechnology may open a broad new avenue and an effective therapeutic strategy to treat various types of human cancers.

## Figures and Tables

**Figure 1 molecules-28-01193-f001:**
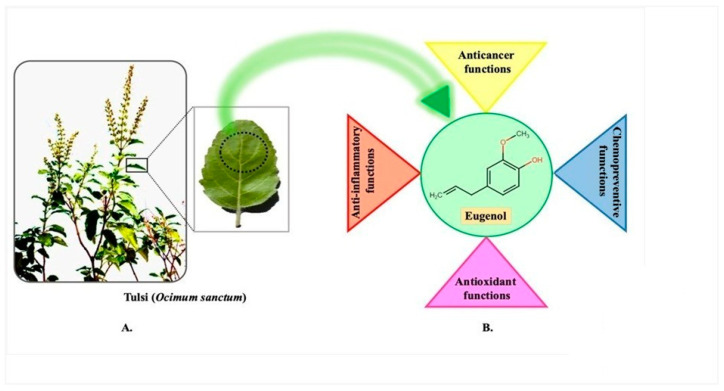
(**A**) Photograph of tulsi (*Ocimum sanctum*) plant and leaf. (**B**) Eugenol’s pharmacological actions include anti-inflammatory, antioxidant, anticancer, and chemopreventive effects.

**Figure 2 molecules-28-01193-f002:**
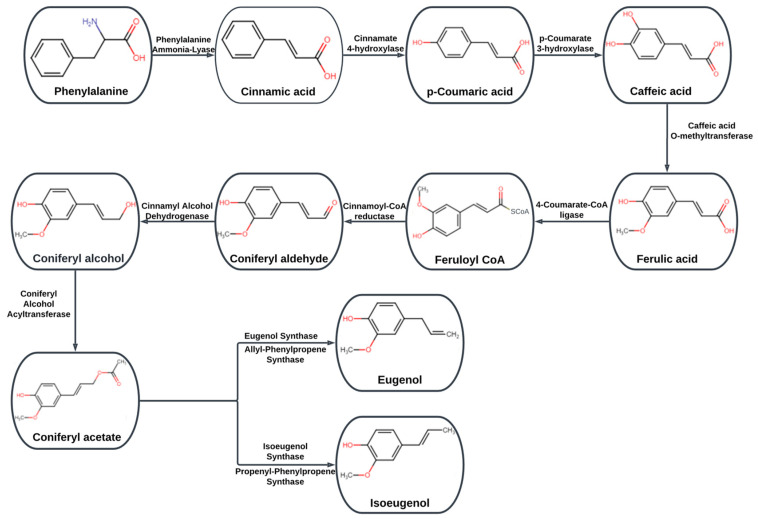
Biosynthetic pathway of eugenol.

**Figure 3 molecules-28-01193-f003:**
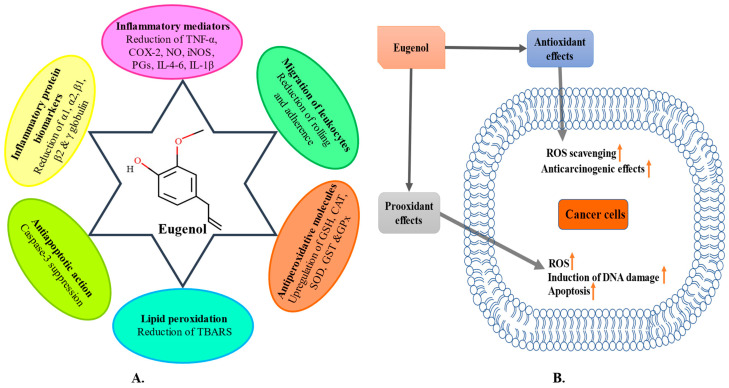
(**A**) Decreased inflammatory response of eugenol with increasing concentration in an animal model or cell culture. (**B**) Antioxidant and prooxidant properties of eugenol.

**Figure 4 molecules-28-01193-f004:**
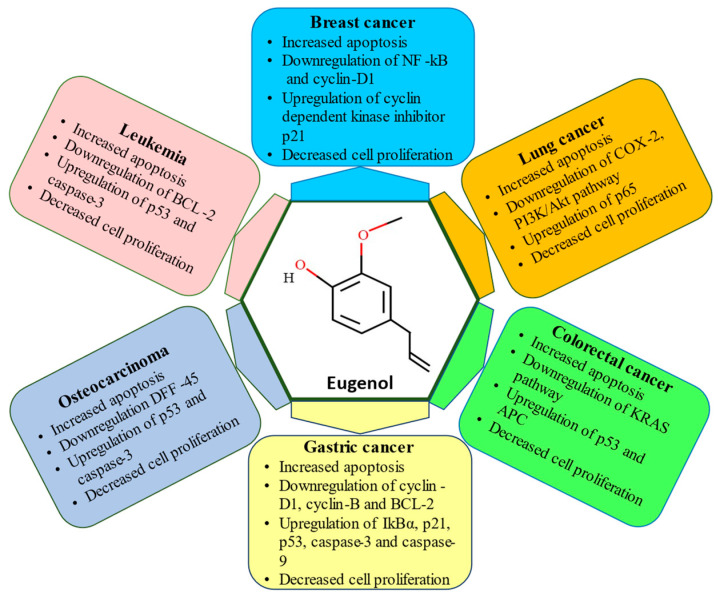
Anti-cancerous effects of eugenol on different types of cancer.

**Table 1 molecules-28-01193-t001:** List of phytoconstituents present in tulsi (*O. sanctum*).

S.No.	Compound Name	IUPAC Name	Chemical Formula	PubChem CID
1.	Eugenol	2-Methoxy-4-prop-2-enylphenol	C_10_H_12_O_2_	3314
2.	Ursolic acid	(1S,2R,4aS,6aR,6aS,6bR,8aR,10S,12aR,14bS)-10-hydroxy-1,2,6a,6b,9,9,12a-heptamethyl-2,3,4,5,6,6a,7,8,8a,10,11,12,13,14b-tetradecahydro-1H-picene-4a-carboxylic acid	C_30_H_48_O_3_	64945
3.	Apigenin	5,7-Dihydroxy-2-(4-hydroxyphenyl)chromen-4-one	C_15_H_10_O_5_	5280443
4.	Caryophyllene	(1R,4E,9S)-4,11,11-trimethyl-8-methylidenebicyclo[7.2.0]undec-4-ene	C_15_H_24_	5281515
5.	Carvacrol	2-Methyl-5-propan-2-ylphenol	C_10_H_14_O	10364
6.	Cirsimaritin	5-Hydroxy-2-(4-hydroxyphenyl)-6,7-dimethoxychromen-4-one	C_17_H_14_O_6_	188323
7.	Estragole	1-Methoxy-4-prop-2-enylbenzene	C_10_H_12_O	8815
8.	Linalool	3,7-Dimethylocta-1,6-dien-3-ol	C_10_H_18_O	6549
9.	Oleanolic acid	(4aS,6aR,6aS,6bR,8aR,10S,12aR,14bS)-10-hydroxy-2,2,6a,6b,9,9,12a-heptamethyl-1,3,4,5,6,6a,7,8,8a,10,11,12,13,14b-tetradecahydropicene-4a-carboxylic acid	C_30_H_48_O_3_	10494
10.	Rosemarinic acid	3-(3,4-Dihydroxyphenyl)-2-[I-3-(3,4-dihydroxyphenyl)prop-2-enoyl]oxypropanoic acid	C_18_H_16_O_8_	5315615

**Table 2 molecules-28-01193-t002:** Nutritive contents of *Ocimum sanctum*(USDA, FDC, 2019)[53].

S. No.	Nutritional Components	Contents (per 100 g)
1.	Protein	3.15 g
2.	Carbohydrate	2.65 g
3.	Fat	0.64 g
4.	Calcium	177 mg
5.	Vitamin C	18 mg
6.	β-Carotene	3140 µg
7.	Copper	0.385 mg
8.	Iron	3.17 mg
9.	Magnesium	64 mg
10.	Phosphorus	56 mg
11.	Zinc	0.81 mg
12.	Sodium	4 mg
13.	Potassium	295 mg

**Table 3 molecules-28-01193-t003:** Eugenol’s molecular processes in relation to its several nutritional benefits.

Bioactive Properties	Molecular Mechanism	References
Chemopreventive activity	Reduces MMP-2 and phosphate-Akt expression in a human lung cancer cell line	[140,141,142,143,144]
	EEOS’ potential use as a chemopreventive for lung cancer	
	Chemopreventive effects of eugenol on stomach cancer caused by N-methyl-N′-nitro-N-nitrosoguanidine (MNNG)	
Anticancer activity	Inhibits the COX-2 gene in human colon HT-29 cell lines	[38,46,48,69,141,145,146,147,148,149]
	Apoptosis in MCF-7 human breast cancer cells and gastric cancer AGS cells	
	Diminished oxidation of DNA	
	Inhibits the action of matrix metalloproteinase (MMP-9)	
	Prevents the synthesis of prostaglandin-E2	
	Triggers cell apoptosis	
	Target surviving/E2F1 pathways	
	Inhibits ERK pathways/proteins	
Anti-inflammatory Activity	Suppression of chemotaxis of neutrophils and macrophages	[69,150,151,152,153]
	Prevents the expression of inflammatory cytokines	
	Inhibitory impact on prostaglandin production	
	Negatively regulates TNF-α	
	Inhibits COX-2 activity	
	Inhibits NF-kappaB pathways	
Antioxidant activity	Suppressive effect on lipid peroxidation	[148,154,155,156,157,158]
	Hinders ROS and RNS production	
	Suppressive effects on hexanal oxidation	
	Inhibitory effects on copper-dependent LDL oxidation	
	Negatively regulates iron-mediated lipid peroxidation	
	Inhibits nonenzymatic peroxidation in liver mitochondria	
	Prevents the emergence of illnesses caused by oxidative stress	

## Data Availability

Data and figures are contained within the manuscript.

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
