# Peer review of "An Update on the Therapeutic Anticancer Potential of *Ocimum sanctum* L.: “Elixir of Life”"

_molecules, 2023, doi:10.3390/molecules28031193_

Round 1
Reviewer 1 Report
Please find the Attached file in PDF for comments to author.

Author Response
Point 1: Reviewer Comment:
The citations reference in Tables should be followed as in body text as instructed by the journals. (143)
Our Response1: Thank you for taking the time to read the manuscript and comment. We have made the necessary corrections.
Point 2: Figure 1 quality need to improve.
Our Response:
Your support and encouragement are greatly appreciated. Improvement were made to Figure1.

Reviewer 2 Report
The work is very interesting. I have some technical complaints at work. You need to know and remember that the names of Families (Lamiaceae, Solanaceae etc.) must write non italic. Italic we write only the Genus and Species names (Ocimum sanctum).
NB !!! BIG MISTAKES - On page 5 - ""Additionally, it has demonstrated antibacterial effects against a variety of human illnesses, including Fasciola gigantica, Giardia lamblia, and Haemonchuscontortus"". Haemonchuscontortus must write separately - Haemonchus contortus.
Fasciola gigantica, Giardia lamblia, and Haemonchus contortusis are parasites, not illnesses! And not bacteria too! The authors confuse the concepts of disease, bacterium, parasite! So how Eugenol has an antibacterial effect against parasites?!? Big mistake and confusion! It's better to remove these information.
Author Response
Comments and Suggestions for Authors
The work is very interesting. I have some technical complaints at work. You need to know and remember that the names of Families (
Comments and Suggestions for Authors
The work is very interesting. I have some technical complaints at work. You need to know and remember that the names of Families (Lamiaceae, Solanaceae etc.) must write non italic. Italic we write only the Genus and Species names (Ocimum sanctum).
Our Response: Thanks for your time and review. This was a typological error, now it is corrected.
NB !!! BIG MISTAKES - On page 5 - ""Additionally, it has demonstrated antibacterial effects against a variety of human illnesses, including Fasciola gigantica, Giardia lamblia, and Haemonchuscontortus"". Haemonchuscontortus must write separately - Haemonchus contortus.
Fasciola gigantica, Giardia lamblia, and Haemonchus contortusis are parasites, not illnesses! And not bacteria too! The authors confuse the concepts of disease, bacterium, parasite! So how Eugenol has an antibacterial effect against parasites?!? Big mistake and confusion! It's better to remove these information.
Our Response: We deeply apologize for the error. Thank you kindly for pointing out the error. It has been eliminated from the manuscript.
etc.) must write non italic. Italic we write only the Genus and Species names (Ocimum sanctum).
Our Response: Thanks for your time and review. This was a typological error, now it is corrected.
NB !!! BIG MISTAKES - On page 5 - ""Additionally, it has demonstrated antibacterial effects against a variety of human illnesses, including Fasciola gigantica, Giardia lamblia, and Haemonchuscontortus"". Haemonchuscontortus must write separately - Haemonchus contortus.
Fasciola gigantica, Giardia lamblia, and Haemonchus contortusis are parasites, not illnesses! And not bacteria too! The authors confuse the concepts of disease, bacterium, parasite! So how Eugenol has an antibacterial effect against parasites?!? Big mistake and confusion! It's better to remove these information.
Our Response: We deeply apologize for the error. Thank you kindly for pointing out the error. It has been eliminated from the manuscript.
Reviewer 3 Report
The manuscript entitled "An update on the therapeutic anticancer potential of Ocimum sanctum L.: "Elixir of Life"" is a good review work on the plant. However, I have some major concerns about the writing part of the review article. Hence, after a major modification of the article it can be considered for publication.
1. The abstract seems to be very short; I suggest to follow the pattern in an unstructured manner like "a brief background on the ethnopharmacology of Ocimum, main objectives of the review, main findings of the review, conlcusion and future perspectives".
2. I think the scientific name "Ocimum sanctum" is a synonym to "Ocimum tenuiflorum" Please check and modify accordingly.
3. The introduction may be started by emphasizing the ethnopharmacological potential of the Ocimum species in general. Later the authors can focus on the specific plant.
4. After the introduction portion (section no 1), next section is on eugenol (section 2). I suggest to provide the section 2 as the phytochemical compositoin. Authors can discuss the chemical components of different Ocimum sanctum extracts and also the Ocimum sanctum essential oils.
5. Authors discussed about the antioxidant and antiinflammatory potentials of Eugenol. Where is the details of antioxidant and antiinflammatory properties of Ocimum? I suggest to include more details on the plant rather than the pure compounds.
6. I suggest to include recent articles such as https://doi.org/10.1016/j.pmpp.2021.101746 (on the ultrasound assisted extraction and anticancer activity), https://doi.org/10.1016/j.pmpp.2021.101759 (cytoprotective effect)
7. Again in Section 6, the authors described about Skin cancer and Eugenol? I feel authors need to change the title as "An update on the therapeutic anticancer potential of Eugenol: "Elixir of Life". It is not funny to give one plant in title and discuss about a minute component present in it in the text.
8. The same comments are applicable in the remaining parts also; authors need to check the manuscript thoroughly.
9. Include recent studies on the plant (not about a single compound).
10. There are several other bioactive compounds present in the Ocimum sanctum. Authors need to include those details in the manuscript.
Author Response
Response to Reviewer 3 Comments
Comments and Suggestions for Authors
The manuscript entitled "An update on the therapeutic anticancer potential of Ocimum sanctum L.: "Elixir of Life"" is a good review work on the plant. However, I have some major concerns about the writing part of the review article. Hence, after a major modification of the article it can be considered for publication.
Reviewer Comments
- The abstract seems to be very short; I suggest to follow the pattern in an unstructured manner like "a brief background on the ethnopharmacology of Ocimum, main objectives of the review, main findings of the review, conlcusion and future perspectives".
Our Response1: We appreciate you taking the time to read our paper and provide feedback. We've updated the abstract to reflect your thoughtful suggestions, which is a good thing to say. The change has been made to the abstract (Red coloured text)
Reviewer Comments
- I think the scientific name "Ocimum sanctum" is a synonym to "Ocimum tenuiflorum" Please check and modify accordingly.
Our Response: Yes, Ocimum tenuiflorum is a synonyms, there is no difference between the two. (http://www.theplantlist.org/tpl/record/kew-137068).
However both names are equally accepted. Tulsi is also known as holy basil, Krishna tulsi, and numerous other names.
Reviewer Comments
- The introduction may be started by emphasizing the ethnopharmacological potential of the Ocimum species in general. Later the authors can focus on the specific plant.
Our Response: We appreciate your insightful suggestion. The change has been made in response to suggestions. We included Ocimum species in the text (Red coloured). Ocimum basilicum and Ocimum tenuiflorum are widely used in cooking and medicine. We in cluded i)O. americanum L. ii) O. basilicum L. iii) O. campechianum Mill., iv)O centraliafricanum, v) (O. gratissimum L.) is native to Africa, vii) Ocimum tenuiflorum L. (O. sanctum).
Reviewer Comments
- After the introduction portion (section no 1), next section is on eugenol (section 2). I suggest to provide the section 2 as the phytochemical compositoin. Authors can discuss the chemical components of different Ocimum sanctum extracts and also the Ocimum sanctum essential oils.
Our Response: We appreciate your valuable suggestion. The suggested change has been implemented.
Reviewer Comments
- Authors discussed about the antioxidant and antiinflammatory potentials of Eugenol. Where is the details of antioxidant and antiinflammatory properties of Ocimum? I suggest to include more details on the plant rather than the pure compounds.
Our Response: We appreciate your valuable suggestion. The modification that was suggested has been carried out and incorporated into the introduction.
- I suggest to include recent articles such as https://doi.org/10.1016/j.pmpp.2021.101746 (on the ultrasound assisted extraction and anticancer activity), https://doi.org/10.1016/j.pmpp.2021.101759 (cytoprotective effect)
- Our Response: We appreciate your valuable suggestion. We incorporated this article.
- Reviewer Comments -Again in Section 6, the authors described about Skin cancer and Eugenol? I feel authors need to change the title as "An update on the therapeutic anticancer potential of Eugenol: "Elixir of Life". It is not funny to give one plant in title and discuss about a minute component present in it in the text.
Our Response: We're sorry about the confusion. The main bioactive compound in tulsi is eugenol, which makes up 70% of the plant. We are now adding other bioctive compounds to the list.
Reviewer Comments
- The same comments are applicable in the remaining parts also; authors need to check the manuscript thoroughly.
Our Response: We appreciate your valuable suggestion. The modification that was suggested has been applied.
Reviewer Comments
- Include recent studies on the plant (not about a single compound).
Our Response: We appreciate your valuable suggestion. We included some of the most recent research on Tulsi, which was published in 2022.
Reviewer Comments
- There are several other bioactive compounds present in the Ocimum sanctum. Authors need to include those details in the manuscript.
Our Response: We appreciate your valuable suggestion. We are now adding other bioctive compounds to the list.
Round 2
Reviewer 3 Report
Authors have improved the manuscript in response to the comments.